# *Ent*-homocyclopiamine B, a Prenylated Indole Alkaloid of Biogenetic Interest from the Endophytic Fungus *Penicillium concentricum*

**DOI:** 10.3390/molecules24020218

**Published:** 2019-01-09

**Authors:** Tehane Ali, Tiffany M. Pham, Kou-San Ju, Harinantenaina L. Rakotondraibe

**Affiliations:** 1Division of Medicinal Chemistry and Pharmacognosy, College of Pharmacy, The Ohio State University, Columbus, OH 43210, USA; tehane.ali@bayer.com (T.A.); ju.109@osu.edu (K.-S.J.); 2Department of Microbiology, College of Arts and Sciences, The Ohio State University, Columbus, OH 43210, USA; pham.274@osu.edu; 3Infectious Diseases Institute, The Ohio State University, Columbus, OH 43210, USA; 4Center for Applied Plant Sciences, The Ohio State University, Columbus, OH 43210, USA

**Keywords:** liverwort, endophyte, fungus, *Penicillium*, metabolite, indole alkaloid, antimicrobial

## Abstract

*Ent*-homocyclopiamine B**** (**1**), a new prenylated indole alkaloid bearing an alicyclic nitro group along with 2-methylbutane-1,2,4-triol (**2**) were isolated from an endophytic fungus *Penicillium concentricum* of the liverwort *Trichocolea tomentella* (Trichocoleaceae). The structure of **1** was elucidated through extensive spectroscopic analyses and comparison with data reported for a structurally related nitro-bearing *Penicillium* metabolite, clopiamine C (**3**), which contain an indolizidine ring instead of the quinolizine ring in **1**. The new compound, *ent*-homocyclopiamine B,**** exhibited slight growth inhibition against Gram-positive bacteria. Based on the reported biosynthesis of related compounds and the isolation of the mevalonic acid derived compound 2-methyl-1,2,4-butanetriol (**2**), we proposed that *ent*-homocylopiamine B (**1**) was biosynthesized from lysine and prenyl group-producing mevalonic pathway.

## 1. Introduction

Endophytes are microorganisms (bacteria and fungi) that live within plants without causing apparent pathogenic disease symptoms. Although researchers are seeking to understand the roles of most endophytes in *planta*, endophytes are believed to protect plant hosts by producing biologically active molecules, reported to secure growth by biosynthesizing plant hormones (e.g., gibberellins), and to balance plant microbial community by counteracting activities of pathogens. It is obvious that plants acquire their endophytes from the environment and thus there should not be any difference between endophytic and non-endophytic microbes of the same species. The only dissimilarity is undoubtedly the surrounding environment where they adapt and acquire starting materials for the biosynthesis of their bioactive metabolites that they use for symbiosis. This adaptation makes endophytes capable to produce wide range of compounds when isolated from the host and fermented using various media. Compounds exhibiting antimicrobial, antiviral and antiproliferative activity have been reported from endophytes [1]. We recently described chemical and pharmacological investigations of extracts prepared from a *Penicillium* (Trichocomaceae) fungus endophytic to the liverwort *Trichocolea tomentella* (Trichocoleaceae) that led to the isolation of the antifungal griseofulvin, dehydrogriseofulvin, dechlorogriseofulvin, dehydrodechlorogriseofulvin, griseophenones B and C, griseoxanthones, driniopsin H, norlichexanthonealternariol, ethylene glycol benzoate, 6-chloro-3,8-dihydroxy-1-methylxanthone, *epi*-epoxydon, gentisyl alcohol, gentisyl quinone, hydroxychlorogentisyl quinone, chlorogentisyl quinone, bromogentisyl alcohol and the identification of compounds exhibiting antiproliferative activity against hormone-dependent breast cancer cell line (MCF-7) and the human colorectal adenocarcinoma cell line HT-29. In the continuation of our systematic studies aiming to understand the metabolic profile of the same endophytic strain and to isolate new and bioactive compounds, we prepared an extract of its fermentation in brown rice supplemented with sodium fluoroacetate and isolated 2-methylbutane-1,2,3-triol (**2**) [2] and *ent*-homocyclopiamine B**** (**1**), a nitro containing alicyclic alkaloid. The isolation and the structure elucidation of the new compound, the effect of the supplementation of sodium monofluoroacetate to the media, the biosynthetic pathway of (**1**) as well as the results of its antimicrobial evaluation will be discussed herein.

## 2. Results

Compound **1** was obtained as white powder and its positive HRESIMS displayed a protonated molecular ion peak at *m*/*z* 482.2643 ([M + H]^+^ which corresponds to a molecular formula of C_27_H_35_N_3_O_5_ (calcd for C_27_H_36_N_3_O_5_^+^, 482.2649). The IR spectrum exhibited two characteristic stretches (1589 and 1353 cm^−1^) of a nitro group [3,4]. The UV maxima (λmax 206, 232, 262, 270 sh, 349 nm) of **1** were very similar to those observed in clopiamine C (**3**), a nitro-containing indole alkaloid derivative isolated from a species of the same genus *Penicillium* (*Penicillium* sp.) [3]. The ^1^H-NMR spectrum displayed four upfielded singlet methyls at δ 0.82, 0.95, 1.40 and 1.7 ppm (each corresponds to 3H), an aromatic methoxyl group (δ 3.80, s, 3H), two methine protons at δ 2.87 (m) and 3.17 (m), three sets of diastereotopic methylene protons at: δ 2.50 and 2.97 (each doublet, H-5ab); δ 3.48 and 3.15 (each doublet, H-20ab); and δ 2.60 and 2.94 (each doublet, H-24ab), five pairs of methylene protons at δ 1.33 (m, H-15a) and 2.77 (dt, H-15b), 1.22 (m, H-19a) and 1.99 (dq, H-19b), δ 1.25 (m, H-18a) and 1.68 (qt, H-18b), δ 1.58 (qt, H-17a) and 1.92 (brd, H-17b), δ 2.83 (brd, H-20a) and 2.86 (m, H-20b); and two *ortho* coupled aromatic protons (δ 6.65 and δ 7.30, each doublet). The ^13^C-NMR spectrum displayed 27 carbon signals corresponding to five methyls (23.5, C-29; 23.7, C-26; 24.4, C-28; 26.8, C-25; and 56.5, C-27), six carbons assignable to a tetrasubstituted aromatic ring (108.4, C-7; 160.2, C-8; 105.2, CH-9; 133.6, CH-10; 122.0, C-11 and 149.9, C-12), four quaternary carbons (62.3, 57.9, 48.8, and 97.2), one of which assignable to a nitro group bearing quaternary carbon, two methines (δ 48.8 and 57.0), eight *sp*^3^ hybridized methylenes (δ 21.1, C-18; 26.1, C-19; 26.9, C-17; 27.2, C-15; 44.7, C-24; 54.1, C-5; 56.0, C-20; and 60.1, C-22), and two carbonyls ascribable to a ketone (192.9, C-6) and an amide carbonyl (δ 183.3, C-2). The ^1^H and ^13^C-NMR data of compound **1** were very similar to those of **3** except for the presence of an additional set of methylene signals arising from the A-ring of **1**. Interpretation of the Correlation Spectroscopy (COSY) spectroscopic data concluded that the additional methylene was part of the spin network of a partial structure –CH-CH_2_-CH-CH_2_-CH_2_-CH_2_-CH_2_- (from H-14 to H-20). In addition, the three observed methylene protons (δ 3.48 and 3.15, each doublet, *J* = 11.9 Hz, H-22ab; δ 2.60, and 2.94, d, *J* = 16.1 Hz, H-24ab; and δ 2.50 and 2.97, d, *J* = 15.6 Hz, H-5ab) of C-22, C24, and C-5 respectively were concluded to be isolated. The allocation of all functionalities present in **1** was carried out by comparison of its spectroscopic data with those of clopiamine C (**3**) as well as interpretation of its HMBC spectroscopic data.

The presence of a 2,2,-dimethyl-2,3-dihydroquinolin-4(1H)-one was confirmed by the HMBC long-range correlations from H-25 and H-26 to C-4 and C-5, H-5 to C-6 and C-7, OCH_3_ to C-8, the ortho-coupled methine proton at δ.65 (H-9) to C-7, C-8, and C-11. The presence of 8,8-dimethyldecahydro-1H-cyclopenta[f]quinolizine in **1** instead of 8,8-dimethyldecahydro-1H-cyclopenta[f]indolizine (**3**) was substantiated by the above-mentioned spin network deduced from the COSY spectrum and the HMBC correlations from H-28/29 to C-1, C-13 and C-14, from H-20 to C-22, from H-22 to C-24, C-14, and C-23. The above data together with the HMBC correlation observed from H-10 to the quaternary carbon at δ 62.3 (C-1) and the presence of 12 degree of unsaturation in **1** allowed us to conclude that as in **3**, the structure of **1** must also contain a spiro-oxindole moiety.

The relative configurations at C-1, C-14, C-16 and C-23 of **1** were determined by NOE experiment while the absolute configurations of **1** were established by comparison of its optical rotation and CD spectroscopic data with those of **3**, the absolute stereostructure of which was deduced by NMR, CD and X-ray diffraction crystallography analyses. The CD spectrum of **1** which displayed negative and positive Cotton effects [(Δε_356_ − 0.92) and (Δε_242_ − 6.08), and (Δε_265_ + 2.82) and (Δε_229_ + 2.8), respectively) is superimposable with those of **3** [3].

Clopiamine C (**3**) belongs to the rare alkyl nitro group-bearing prenylated spirooxindole alkaloids of *Penicillium* species. Of this group of compounds, only cyclopiamine and citranilin-type have been reported (Figure 1) [3,4,5,6]. It is worthy to note that clopiamine C (**3**) and cyclopiamine B (Figure 1) are enantiomers. Their NMR spectroscopic data (although measured in different solvent) are superimposable while their reported optical rotations ([α]_D_) have opposite sign (−97.9° and +117, respectively) [3,4]. From these data, the absolute configurations at C-1, C-14, C-16, and C-22 were thus determined to be *S*, *R*, *S*, and *R*, respectively and the structure of **1** was deduced to be *ent*-homocyclopiamine B as depicted in Figure 2.

The isolation of **1** from *Penicillium concentricum* adds to the list of *Penicillium* strains that produce cyclopiamines. To determine if supplementation of sodium monofluoroacetate to the rice medium affects production of **1**, an ethyl acetate extract of *P. concentricum* grown on brown rice medium and an ethyl acetate extract of the fungus fermented on brown rice supplemented with sodium monofluoroacetate were analyzed using LC-MS. *Ent*-homocyclopiamine B and its derivatives have been detected in the LCMS spectra of both extracts showing that the supplementation did not influence the production of compound **1**. This was confirmed by positive ion electrospray LC-MS screening of the two extracts, which both displayed the peak at *m*/*z* 482.2650 (compound **1**). Additionally, LC-MS analyses (Appendix A) of the extracts also showed a peak corresponding to the characteristic loss of a nitro-radical (*m*/*z* 435.2642) from the molecular ion of **1** [3].

The possible of biosynthesis of citranilins and cyclopiamines have been studied using stable isotope (^13^C) labelling experiments to conclude glucose, anthranilic acid and ornithine as proposed precursors [6]. The present investigation led to the isolation of mevalonic acid derived compound 2-methylbutane-1,2,3-triol (**2**), suggesting that the prenyl group in **1** may be introduced from the catabolism of glucose through mevalonic pathway. Lysine instead ornithine was may be involved in the biosynthesis of the indolizidine component of **1**, analogous to ornithine in clopiamine C. The clear observation of three units of isoprene and a lysine residue in the structure of **1** suggests that further investigation is needed to understand if an aminophenol related compound (apart from the reported anthranilic acid) may be involved in its biosynthesis (Figure 3).

Next, the ethyl acetate fraction was subjected to fluorine NMR experiments (^1^H coupled and decoupled experiments). As results, the presence of fluorinated compounds was clearly observed in the two experiments performed (Data not shown). However, fluorinated compounds were not isolated during this study due to their apparent instability of during the isolation process.

### Antibacterial Activity of Ent-homocyclopiamine B

Prenylated indole alkaloids have been reported to wide-ranging bioactivities including antibacterial, cytotoxic, and insecticidal properties. The antibacterial activity of **1** was tested against a panel of Gram-positive and -negative strains. Although **1** inhibited growth of *Bacillus subtilis* ATCC 6633 and *Mycobacterium smegmatis* NRRL B-14646 on agar plates, zones were significantly smaller than from an equivalent amount of kanamycin (Appendix A). Compound **1** showed no activity against *Escherichia coli* K12, *Salmonella* LT2, *Micrococcus luteus* ATCC 4698, *Pseudomonas putida* PRS2000, and *Serratia marcescens* NRRL B-2544 in this assay.

*B. subtilis* ATCC 6633, *Rhodococcus jhostii* RHA1, and *Corynebacterium glutamicum* NRRL B-2784 were partially inhibited in microbroth dilution assays with 100 μM of **1** (30% inhibition compared with background controls), but not with any concentrations below. No detectable activity was observed against any of the other tested strains.

In summary, inhibition was observed against some (but not all) of the tested Gram-positive strains. None of the tested Gram-negative bacteria were susceptible to **1**.

## 3. Materials and Methods

^1^H and ^13^C-NMR spectra were recorded at 25 °C with a Bruker Avance III 400 HD NMR spectrometer and Bruker Avance III HD Ascend 700 MHz. High-resolution mass spectra were acquired with a Thermo LTQ Orbitrap, specifications; analyzer: ITMS and FTMS, mass range: 50–4000 *m*/*z*, resolution: 7500–100,000. Optical rotations were determined on an Anton Paar MCP 150 polarimeter. Ultraviolet spectra (UV) were recorded using a Hitachi U-2910 UV/Vis double-beam spectrophotometer (Hitachi High- Technologies America, Schaumburg, IL, USA). Circular dichroism was measured on a Jasco J-810 spectropolarimeter. Thermo LTQ Orbitrap and Agilent 1100 HPLC were used to measure LC-MS spectra.

### 3.1. Fungal Source

*Penicillium concentricum*, the endophytic fungus used in this study was isolated from healthy liverwort *Trichocolea tomentella* and identified as previously described [1].

### 3.2. Solvent Extraction and Partition of Fungal Material Fermented on Rice Medium Supplemented with Sodiummonofluoroacetate

Eleven-days fungal culture in rice medium supplemented with sodium monofluoroacetate was soaked with (2 × 3 L) EtOAc and left for overnight at room temperature. The EtOAc extract was evaporated under vacuum to yield a greenish residue (2.3 g). The extract residue was dissolved in methanol and partitioned with hexane (3 × 250 mL) to afford 1.1 g of hexanes and 367 mg of methanol fractions. The MeOH fraction was fractionated on C_18_ reversed-phase silica gel liquid chromatography using 40% aqueous methanol (150 mL) followed by 70% aqueous methanol (150 mL) and later 100% methanol to yield three fractions (B2-1, 127 mg), (B2-2, 78 mg) and (B2-3, 51 mg). B2-1 (127 mg) was subjected to Silica gel column chromatography (2.2 cm × 28 cm; solvent system: 15:6:1 (CHCl_3_: MeOH: H_2_O)) to give six sub-fractions B2-1-1 through B2-1-6. Fraction B2-1-1 and B2-1-4 yielded compound **1** (0.93 mg) and compound **2** (3.7 mg), respectively.

### 3.3. LC-MS Method

LC-MS experiment was performed using Agilent 1100 HPLC binary and Thermo LTQ Orbitrap Mass spectrometer. For the HPLC, Beckman ODS column (5 μm, 4.6 mm, 25 cm, Part#235329) heated at 25 °C (oven temperature). Sample was injected to a water (containing 0.1% formate)-Acetonitrile (containing 0.1% formate) gradient from 5% to 95% (*v*/*v*) over 20 min with a 5 min hold at 95% (*v*/*v*) for 5 min. The column was then reequilibrated to 5% (*v*/*v*) for 3 min. The flow rate was kept at 1 mL/min. Mass spectrometry aquisitions were performed using instrument settings as described in Appendix A.

*Ent*-homocyclopiamine B (**1**): White powder, [α]_D_^25^ −22.7 (c 0.1, MeOH); CD (c 0.8, MeOH) ∆ε (nm) +2.8 (229), −6.08 (242), +2.82 (265), −0.92 (356); IR (NaCl, thin film) ν_max_: 2990, 2932, 1706, 1614, 1589, 1353, 1221, 1139; ^1^H-NMR and ^13^C-NMR spectral data, see Table 1; positive HRESIMS *m*/*z* 482.2643 ([M + H]^+^ which corresponds to a molecular formula of C_27_H_35_N_3_O_5_ (calcd for C_27_H_36_N_3_O_5_^+^, 482.2649).

2-methyl-1,2,4-butanetriol (**2**): Colorless oil; ^1^H-NMR (400 MHz, CD_3_OD) *δ* 1.19 (3H, s, H-5), 1.76 (2H, dddd, *J* = 6.5 Hz, 13.9 Hz, 6.8 Hz, H-3), 3.39 (2H, dd, *J* = 15.5 Hz, H-1), 3.39 (2H, dddd, *J* = 15.5 Hz, 6.8 Hz, 6.5 Hz, H-4), HRESIMS *m*/*z* 143.0678 (calcd for C_5_H_12_NaO_3_^+^, 143.0684).

### 3.4. Antibacterial Testing

Assayed strains included *Escherichia coli* K12, *Salmonella enterica* serovar Typhimurium LT2, *Pseudomonas putida* PRS2000, *Serratia marcescens* NRRL B-2544, *Bacillus subtilis* ATCC 6633, *Micrococcus luteus* ATCC 4698, *Mycobacterium smegmatis* NRRL B-14616, *Corynebacterium glutamicum* NRRL B-2784, and *Rhodococcus jhostii* RHA1. Strains PRS2000, ATCC 4698 and RHA1 were grown on nutrient broth (5 g peptone, 3 g meat extract, per L) at 30 °C and all others on LB (5 g NaCl, 5 g yeast extract, 10 g tryptone, per L) at 37 °C. Bacto agar (16 g L^−1^) was added for plates.

Growth inhibition was evaluated on plates inoculated with lawns of the test strains. 50 μL of an overnight culture of bacteria was added to 5 mL of broth and grown until the culture reached a turbidity equal to 1 × 10^7^–2 × 10^8^ CFU/mL (based on a previously determined calibration curve). 100 μL was then evenly spread onto plates using sterile glass beads to dryness. 50 nmoles *ent*-homocyclopiamine B (dissolved in 5 μL 25% DMSO), 50 nmoles kanamycin (dissolved in 5 μL water), or diluent controls (5 μL of sterile water or 5 μL 25% DMSO) were applied, allowed to dry, and then incubated for 16–20 h at the growth temperature appropriate for each strain.

The broth microdilution assay was performed as outlined by the Clinical and Laboratory Standards Institute (CLSI) [7]. Briefly, 2 μL of a 50X stock of *ent*-homocyclopiamine B (dissolved in 25% DMSO) was added to wells of a 96-well plate. Live and background controls received 2 μL of diluent. 88 μL of broth was added to all wells except background control wells which received 98 μL. 50 μL of an overnight culture of bacteria was added to 5 mL of broth and grown until the culture reached a turbidity equal to 1 × 10^7^–2 × 10^8^ CFU/mL (based on a previously determined calibration curve). The culture was diluted to 5 × 10^6^ CFU/mL and 10 μL added to each well except background controls to a final concentration of 5 × 10^5^ CFU/mL. Plates were incubated at 30 °C or 37 °C (as appropriate for each strain) for 16–20 h. Absorbance was then recorded on a Bio-Rad xMark microplate spectrophotometer at λ = 600nm. Minimal inhibitory concentration (MIC) values were defined as the lowest concentration resulting in ≥ 90% growth inhibition.

## Figures and Tables

**Figure 1 molecules-24-00218-f001:**
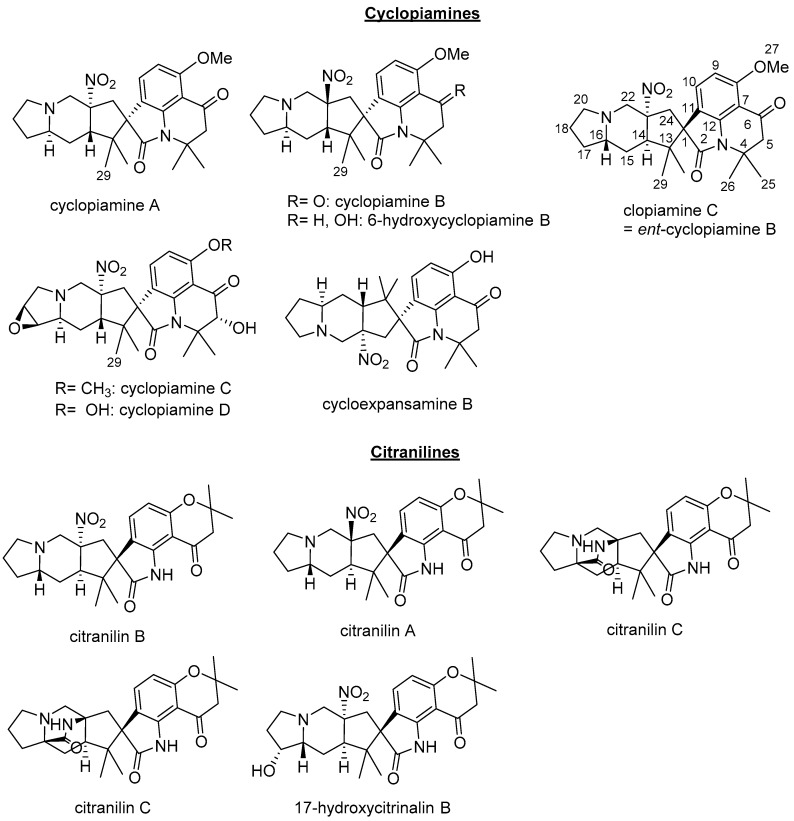
Structures of Cyclopiamines and Citranilines.

**Figure 2 molecules-24-00218-f002:**
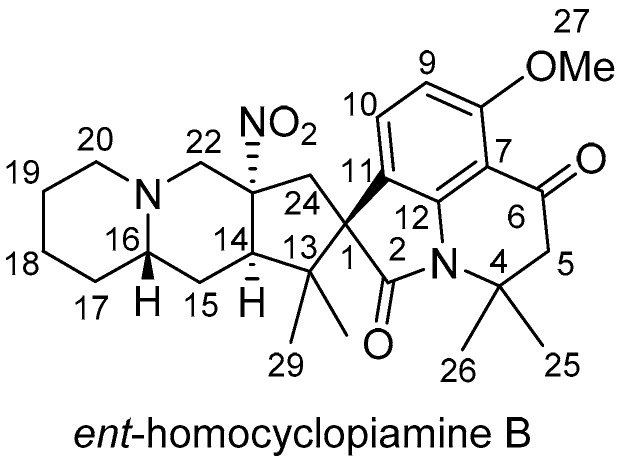
Structure of compound **1.**

**Figure 3 molecules-24-00218-f003:**
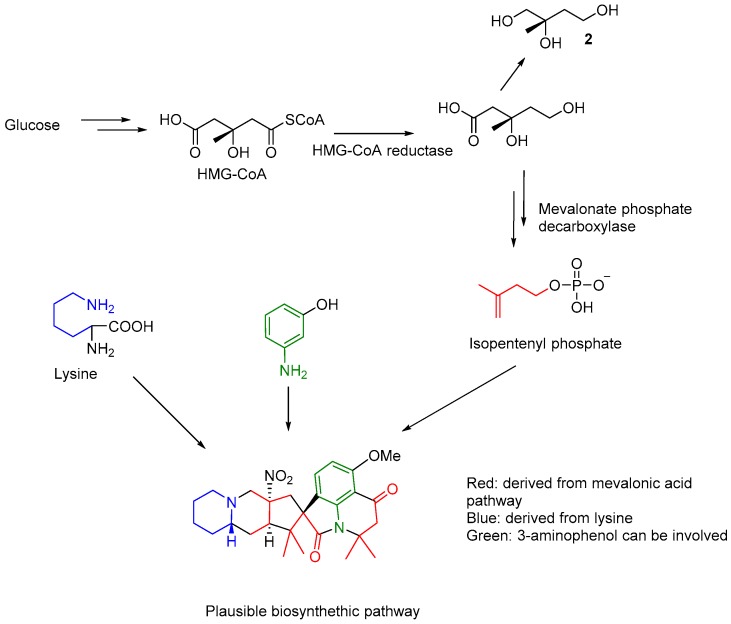
Plausible biosynthetic routes and components of **1.**

**Table 1 molecules-24-00218-t001:** NMR spectroscopic data for compounds **1** and **3**.

Position	*Ent*-homocyclopiamine B (1) *^a^*		Clopiamine C (3) *^b^*	
	^1^H	^13^C	^1^H	^13^C
	δ_H_ (*J* in Hz)	δ_C_, Type	δ_H_ (*J* in Hz)	δ_C_, Type
1		62.3, C		60.2, C
2		183.3, C		180.2, C
3				
4		57.9, C		56.0, C
5	2.50 (d, 15.6) 2.97 (d, 15.6)	54.1, CH_2_	2.87 (d, 15.0) 2.42 (d, 15.0)	52.4, CH_2_
6		192.9, C		189.4, C
7		108.4, C		106.7, C
8		160.2, C		158.4, C
9	6.65 (d, 8.4)	105.2, CH	6.64 (d, 8.4)	103.7, CH
10	7.30 (d, 8.4)	133.6, CH	7.56 (d, 8.4)	132.0, C
11		122.0, C		118.0, C
12		149.9, C		148.0, C
13		49.2, C		48.8, C
14	2.87 (m)	48.8, CH	3.54 (d, 9.0)	44.0, CH
15	1.33 (m) 2.77 (td, 12.9, 6.2)	27.2, CH_2_	1.85 (m) 1.71 (m)	26.8, CH_2_
16	3.17 (m)	57.0, CH	1.92 (m)	61.2, CH
17	1.58 (qt, 12.9, 3.9) 1.92 (br d, 12.5)	26.9, CH_2_	1.90 (m) 1.21 (m)	30.9, CH_2_
18	1.25 (m) 1.68 (qt, 12.9, 3.9)	21.1, CH_2_	1.63 (m)	20.7, CH_2_
19	1.22 (m) 1.99 (dq, 4.0, 12.6)	26.1, CH_2_	2.87 (m) 1.97 (m)	52.9, CH_2_
20	2.83 (br d, 12.5) 2.86 (m)	56.0, CH_2_		
21			3.64 (d, 12.6) 2.70 (d, 12.6)	64.0, CH_2_
22	3.48 (d, 11.9) 3.15 (d, 11.9)	60.1, CH_2_		94.6, C
23		97.2, C	2.68 (d, 15.6) 2.61 (d, 15.6)	41.3, CH_2_
24	2.60 (d, 16.1) 2.94 (d, 16.1)	44.7, CH_2_	1.57 (s)	25.6, CH_3_
25	1.70 (s)	26.8, CH_3_	1.26 (s)	23.2, CH_3_
26	1.40 (s)	23.7, CH_3_	3.79 (s)	55.8, CH_3_
27	3.80 (s)	56.5, CH_3_	0.97 (s)	22.8, CH_3_
28	0.95 (s)	24.4, CH_3_	0.87 (s)	22.8, CH_3_
29	0.82 (s)	23.5, CH_3_		

^a^ 700 MHz for ^1^H-NMR and 150 MHz for ^13^C, measured in CD_3_OD-*d*_4_, ^b^ From reference [3].

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
