# Peer review of "Ent-homocyclopiamine B, a Prenylated Indole Alkaloid of Biogenetic Interest from the Endophytic Fungus Penicillium concentricum"

_molecules, 2019, doi:10.3390/molecules24020218_

Reviewer 1 Report

This article describes a new derivative, Ent-homecyclopiamine B, from the endophytic fungus P. concentricum. This compound features to possess a quinolizine group instead of common indolizine  in many cylcopiamine class. I think that this compound is valuable to report, but this article has some error. And the bioactivity of 1 is not good. The authors need to be compared or evaluated with the bioactivity for the reported cyclopiamine class. Furthermore, the supplementary material is not given in the review site.

The error I found are listed below 

 Line 24, homoclopiamine C (11) ? if yes, here is the structure in your article? Maybe Ent-homocyclopiamine B (1)

Lines 40-44, I think any references should be added after the sentence.

Lines 66-69. I think the coupling constants expressions are not needed because the values are given in Table 1. Somewhat they are confusing.

Line 88, not H-28,29, but H-25/26 

Lines 96 and 107, not C-22, but C-23

Line 97, not configurations of 2, but ~ of 1

Line 178, (c0.1, solvent?) Write the right solvent

Author Response

Editor-in-Chief: of the journal Molecules

Dear Editor

We thank you and the reviewers for the valuable comments. After considering all of the comments we are pleased to submit the revised version of our manuscript entitled “Ent-homocyclopiamine B, a prenylated indole alkaloid of biogenetic interest from the endophytic fungus Penicillium concentricum

by Tehane Ali, Tiffany Pham, Ju, Kou-San, and  I for your consideration for publication in the Special issue: Plant associated Microbes as sources of New Pharmacophores of the Journal Molecules.

Revisions are highlighted.

Comments and Suggestions for Authors

 Reviewer 1

Comment 1: 

This article describes a new derivative, Ent-homecyclopiamine B, from the endophytic fungus P. concentricum. This compound features to possess a quinolizine group instead of common indolizine  in many cylcopiamine class. I think that this compound is valuable to report, but this article has some error. And the bioactivity of 1 is not good. The authors need to be compared or evaluated with the bioactivity for the reported cyclopiamine class. Furthermore, the supplementary material is not given in the review site.

Response:
We agree with the reviewer that the bioactivity of compound 1 is not good. Noteworthy, this is not only for the antimicrobial activity of compound 1 but also for all compounds of this type. Therefore we did not compare the activity of 1 with other cyclopiamine class since none are very active.  

The supporting information was prepared but has not been submitted since it was not included in the required documents in the submission. We are now attaching the supporting information.
The error I found are listed below  

Comment 2:
Line 24, homoclopiamine C (11) ? if yes, here is the structure in your article? Maybe Enthomocyclopiamine B (1)

Response:
We apologize, it should be compound 1. Thus revised 

Comment 3:

Lines 40-44, I think any references should be added after the sentence.
Response:
A reference was added

Comment 4:
Lines 66-69. I think the coupling constants expressions are not needed because the values are given in Table 1. Somewhat they are confusing.
Response:
We agree with the referee. The coupling constants were removed.

Comment 5:
Line 88, not H-28,29, but H-25/26  
Response:
Revised

Comment 6:
Lines 96 and 107, not C-22, but C-23
Response:
Revised

Comment 7:
Line 97, not configurations of 2, but ~ of 1
Response:
Revised

Comment 8: Line 178, (c0.1, solvent?) Write the right solvent    

Response:
Revised

Reviewer 2 Report

Manuscript entitled "Ent-homocyclopiamine B, a prenylated indole alkaloid of biogenetic interest from the endophytic fungus Penicillium concentricum" is within the scope of Molecules. It is good experimental article with interesting subject and good experimental work. The manuscript is fairly well written and includes a great deal of  information. The methodology of experimental part is well established and it does not raise any objections. The results and discussion are represented in a logical way. Authors included figures which are clear and legible.

Just one query has been raised:

Section Material and Methods: Please provide the more information about the LC-MS method used for the identification of compounds.

Author Response

Editor-in-Chief: of the journal Molecules

Dear Editor

We thank you and the reviewers for the valuable comments. After considering all of the comments we are pleased to submit the revised version of our manuscript entitled “Ent-homocyclopiamine B, a prenylated indole alkaloid of biogenetic interest from the endophytic fungus Penicillium concentricum

by Tehane Ali, Tiffany Pham, Ju, Kou-San, and  I for your consideration for publication in the Special issue: Plant associated Microbes as sources of New Pharmacophores of the Journal Molecules.

Revisions are highlighted.

Information about the LC-MS was added

Thank you for your consideration of this manuscript; I look forward to hearing from you once your evaluation is complete.

Round  2

Reviewer 1 Report

Line 183, not 5 mm, but 5 μm

Line 186, I for HPLC ?